# Influence of Barrier Layers on ZrCoCe Getter Film Performance

**DOI:** 10.3390/ma16072916

**Published:** 2023-04-06

**Authors:** Xin Shi, Yuhua Xiong, Huating Wu

**Affiliations:** 1State Key Laboratory for Advanced Materials for Smart Sensing, GRINM Group Co., Ltd., Beijing 100088, China; 2GRIMAT Engineering Institute Co., Ltd., Beijing 101407, China; 3General Research Institute for Nonferrous Metals, Beijing 100088, China

**Keywords:** getter film, ZrCoCe, barrier layers, Ge substrate

## Abstract

Improving the vacuum degree inside the vacuum device is vital to the performance and lifespan of the vacuum device. The influence of the Ti and ZrCoCe barrier layers on the performance of ZrCoCe getter films, including sorption performance, anti-vibration performance, and binding force between the ZrCoCe getter film and the Ge substrate were investigated. In this study, the Ti and ZrCoCe barrier layers were deposited between the ZrCoCe getter films and Ge substrates. The microtopographies of barrier layers and the ZrCoCe getter film were analyzed using scanning electron microscopes. The sorption performance was evaluated using the constant-pressure method. The surface roughness of the barrier layers and the getter films was analyzed via atomic force microscopy. The binding force was measured using a nanoscratch tester. The anti-vibration performance was examined using a vibration test bench. The characterization results revealed that the Ti barrier layer significantly improved the sorption performance of the ZrCoCe getter film. When the barrier material was changed from ZrCoCe to Ti, the initial sorption speed of the ZrCoCe getter film increased from 141 to 176 cm^3^·s^−1^·cm^−2^, and the sorption quantity increased from 223 to 289 Pa·cm^3^·cm^−2^ in 2 h. The binding force between the Ge substrate and the ZrCoCe getter film with the Ti barrier layer was 171 mN, whereas that with the ZrCoCe barrier layer was 154 mN. The results showed that the Ti barrier layer significantly enhanced the sorption performance and binding force between the ZrCoCe getter film and the Ge substrate, which improved the internal vacuum level and the stability of the microelectromechanical system vacuum devices.

## 1. Introduction

Microelectromechanical system (MEMS) devices have been used in aerospace [1], biomedical [2], automotive [3], communications [4], as well as other high-tech fields [5,6], many of which use Ge as a substrate [7,8,9]. Getter films have attracted significant attention for their sorption performance in maintaining and improving the vacuum degree in MEMS vacuum devices for extended periods [10,11,12,13]. Vacuum package and long-term reliability are important in MEMS gyroscopes to ensure their detection accuracy [12,14,15]. The vacuum encapsulation can provide a high-quality factor and reduce noise interference in MEMS accelerators [16]. Vacuum MEMS devices not only require a high degree of internal vacuum, but also long-term internal stability. Thus, it is extremely significant to improve the sorption performance and reliability of the getter films.

ZrCoCe getter films are commonly used, owing to their low activation temperature, good sorption performance [17], compatibility with the MEMS processes [18], and environmental friendliness [19]. A comparison of different gettering film materials is shown in Table 1. However, during the activation, the substrate releases active gas due to heating. If the getter film absorbs the active gas, part of the sorption quantity will be consumed. Bu et al. [20] found that depositing a dense ZrCoCe barrier layer between the ZrCoCe getter film with a columnar structure and the substrate could effectively eliminate the poisoning effect of the active gas released by the substrate on the ZrCoCe getter film. However, with the development of vacuum devices, higher degrees of internal vacuum are required. Therefore, it is necessary to find a new barrier material to improve the sorption performance of the ZrCoCe getter films. For instance, Ti getter films have been widely used because of their good sorption performance and low activation temperature [21,22,23]. The densification of the Ti film can reduce the diffusion channel of active gas within, making Ti film an effective barrier layer. It is postulated that using Ti and ZrCoCe to deposit dense barrier layers can effectively prevent the diffusion of the active gas released by the substrate into the ZrCoCe getter film.

In this study, Ti and ZrCoCe were deposited as barrier layers for getter films. The Ti was selected as the barrier layer material because of its sorption performance [22,24], whereas the ZrCoCe was chosen as the barrier layer material to simplify the process. The sorption performance, binding force, and anti-vibration properties of the ZrCoCe getter films with different barrier layer materials were studied.

## 2. Materials and Methods

### 2.1. Substrate Outgassing Test

A quadrupole mass spectrometer (HIDEN HMT 101, Michigan, US) was used to measure the gas species released by the Ge substrate at 350 °C for 30 min.

### 2.2. Fabrication of Barrier Layers and ZrCoCe Films

A 2-inch, single-crystal Ge wafer with a thickness of 1000 μm was selected as the substrate. ZrCoCe and Ti barrier layers of 50 nm thickness were deposited on the Ge substrates, forming two substrates of (A) ZrCoCe/Ge and (B) Ti/Ge.

After depositing the barrier layer, a 2000 nm ZrCoCe getter film was deposited on A and B. A 2050 nm ZrCoCe getter film was also deposited on a new Ge substrate. The process parameters for magnetron sputtering are listed in Table 2. The structure of the ZrCoCe getter film is illustrated in Figure 1.

### 2.3. Characterization

The process of regaining a clean surface by heating the ZrCoCe getter film is known as activation. The activation parameters for the ZrCoCe getter films were 30 min at 350 °C, in which the temperature was measured using a K-type spot-welded thermocouple attached to the sample.

The sorption performance of the fabricated ZrCoCe getter films was tested using the constant-pressure method. The constant-pressure method inhalation performance test bench was independently developed by Beijing Nonferrous Metals (Beijing, China).

Scanning electron microscopy (SEM) was used to characterize the surface and cross-sectional morphologies of the ZrCoCe getter films with different barrier layers using a JSM-7900F scanning electron microscope (JEOL, Akishima, Japan) with a probe current of 7 A and an acceleration voltage of 5 kV.

Energy Dispersive Spectroscopy (EDS) detects various elements with different characteristic X-ray wavelengths for the purpose of elemental analysis. The composition of the getter films was analyzed by EDS.

The surface roughness of the barrier layer and the getter films was analyzed by Atomic Force Microscopy (AFM Dimension Icon, Bruker, Billerica, MA, USA).

Nanoindentation was used to analyze the binding force between the Ge substrate and the getter films using a TI 900 Tribolndenter (Hysitron, Billerica, MA, USA) nanoindentation loaded in the range of 0–350 mN.

A shock-impact tester (ECON VT-9208, Zhejiang, China) was used to test the anti-vibration and anti-shock properties of the getter films with the different barrier layers. The getter films were subjected to an environment with a peak amplitude of 1.5 mm and an acceleration speed of 9 g, with spanned frequencies of 15–55 Hz [25].

## 3. Results and Discussion

### 3.1. Surface and Cross-Sectional Micromorphologies of Barrier Layers and Getter Films

#### 3.1.1. Surface and Cross-Sectional SEM Images of ZrCoCe and Ti Barrier Layers

The surface and cross-sectional micromorphologies of the ZrCoCe and Ti barrier layers are shown in Figure 2a–d, respectively. The 1μm barrier films were deposited onto the Ge substrates with the magnetron sputtering process parameters of the barrier layer in Table 1 in order to observe the surface and cross-section morphologies of the barrier layers. Figure 3 shows the surface roughness of the barrier layers and the getter films deposited on the different barrier layers. Compared with the Ti barrier layers, it can be seen that the surface of the ZrCoCe barrier layer was very smooth, which was not conducive to the growth of the columnar ZrCoCe getter film. A rough surface is more conducive to the growth of the getter films with larger diameter columnar structures [20], which can improve the sorption performance of the ZrCoCe getter films. Studies by Benvenuti et al. [26] and Xu et al. [27] have shown that a getter film with a porous columnar structure exhibits better sorption performance. The barrier layer with the columnar structure cannot effectively prevent the active gas released by the substrate from diffusing to the getter layer; therefore, the sorption performance of the getter film was mildly affected. Figure 2d shows that the cross-section of the Ti barrier layer grew in a disorderly pattern, which effectively prevented the active gas released by the substrate from diffusing to the getter layer.

#### 3.1.2. Surface and Cross-Sectional Elements and Micromorphologies of ZrCoCe Getter Films

Several Energy Dispersive Spectroscopy images of the ZrCoCe getter film are shown in Figure 4. According to Figure 4, the getter film prepared in this work was Zr_74_Co_20_Ce_6_, and the element was well distributed. The surface and cross-sectional micromorphologies of the ZrCoCe/ZrCoCe/Ge (getter film/barrier layer/substrate) and the ZrCoCe/Ti/Ge are shown in Figure 5. The surface of the ZrCoCe/Ti/Ge was flat, and the diameter of the columnar structure was continuous. A rough surface and a continuous columnar structure may be beneficial for improving the sorption performance of the ZrCoCe getter films. The rough surface of the ZrCoCe getter film had more active sites, and the continuous columnar structure was more conducive to the diffusion of active gasses in the getter film [28]. Therefore, both the rough surface and the continuous columnar structure may have had positive effects on the sorption performance of the ZrCoCe getter films.

### 3.2. Sorption Performance of the ZrCoCe Getter Films with Different Barrier Layers

Based on the literature, the Ge substrates released H_2_, CO, H_2_O, and CO_2_ during the heating process [29]. In this study, the hydrogen sorption performance of the fabricated ZrCoCe getter films was investigated. Figure 6 shows the sorption performance of the ZrCoCe getter films with different barrier layers. Clearly, compared with the ZrCoCe getter film without a barrier layer, the sorption performance of the film with a barrier layer was improved. The sorption performance of the ZrCoCe getter film with the Ti barrier layer was approximately twice that of the ZrCoCe getter film without a barrier layer, and much greater than that of the ZrCoCe getter film with the ZrCoCe barrier layer. The data collected on the sorption performance of the ZrCoCe getter films with different barrier layers and the ZrCoCe getter films [28] are shown in Table 3. Based on these results, the sorption performance of the ZrCoCe getter film with the Ti barrier layer was the most effective solution. This was mainly due to the dense microstructure of the Ti barrier layer, which effectively prevented the active gas released by the substrate from diffusing into the interior of the ZrCoCe getter film. The Ti also acted as an effective getter film.

### 3.3. Sorption Performance of ZrCoCe Getter Films with Ti Barrier Layers

According to the previous results, it can be found that the ZrCoCe getter film with the Ti barrier layer exhibits the best sorption performance. The diffusion distance of the active gas released from the Ge substrate to the getter film and the residual impurity gas that finally reaches the getter film is different because of the influence of the barrier layer thickness. Figure 7 and Figure 8a show the cross-sectional micromorphology and the sorption performance of the ZrCoCe getter films with different thicknesses of Ti barrier layers. It should be noted that, regardless of the thickness of the barrier layer, the total thickness of the barrier layer and the getter film is about 2.1 μm. It can be seen that thicker Ti barrier layers lead to better sorption performance of the ZrCoCe getter films. Figure 8b shows the sorption performance of the ZrCoCe getter films with the 240 nm and 400 nm Ti barrier layers. The detailed sorption performance data are shown in Table 4. With an increase in the thickness of the Ti barrier layer, the sorption performance of the ZrCoCe getter film also increased due to the remarkable sorption performance and the protection property of the Ti barrier layer.

### 3.4. The Binding Force of ZrCoCe Getter Films with Different Barrier Layers

After the vibration test based on the parameters mentioned above, no small particles or scratches were observed on the surface of the getter films under a low-power microscope. SEM was used for further analysis. The SEM images of the ZrCoCe getter films with different barrier layers after vibration and activation are shown in Figure 9. Figure 9a,c show SEM images after the vibration test, which are useful for proving the anti-vibration properties of the getter films, whereas the SEM images of the getter films after activation are shown in Figure 9b,d. Clearly, no small particles or defects were observed, which correlates well with the evidence from Choa, S. H. [25]. As shown in Figure 9b,d, cracks caused by activation were observed on the surface of the getter films [27]. Compared with the Ti barrier layer, the ZrCoCe getter film with the ZrCoCe barrier layer exhibited more cracks that were similar to the cracks on the surface after activation. Although those cracks may have had no impact on the sorption performance of the ZrCoCe getter film, they may have reduced the binding force between the getter film and the Ge substrate. The analysis of the anti-vibration performance shows that the combination of the Ge with the ZrCoCe/Ti getter film results in a strong structure, which ensures that the getter film can withstand vibration impact to a certain extent.

A strong binding force was necessary for the ZrCoCe getter films to withstand the vibration of the MEMS devices. Figure 10 shows the binding force analysis results for the ZrCoCe getter films prior to activation. The binding force test results obtained using the nanoindenter are presented in Table 5, and the binding force between the ZrCoCe getter film with the Ti barrier layer (ZrCoCe/Ti) and the Ge substrate peaked at 171 mN. The binding force was significantly enhanced by the Ti barrier layer. The binding force between the ZrCoCe getter films and the Ge substrates is the most critical index for evaluating the mechanical properties of the getter film, which is important for the reliability of the ZrCoCe getter films. Small particles were not observed after scratching by the probe, which leads to difficulties in the measurement of vacuum devices and short circuits [30].

## 4. Conclusions

Both Ti and ZrCoCe were deposited as barrier layers between the ZrCoCe getter films and the Ge substrates. The barrier layers improved the sorption performance of the ZrCoCe getter films and increased the binding force between the ZrCoCe getter films and the Ge substrates. The Ti barrier layer was more effective than the ZrCoCe barrier layer for improving the sorption performance of the ZrCoCe getter film because the Ti barrier layer had a rough surface and a dense internal structure. Accordingly, the thicker the Ti barrier layer, the better the sorption performance of the ZrCoCe getter film. The thickness of the Ti barrier layer depended on the space inside the vacuum device. The initial sorption speed of the ZrCoCe getter film with a 50 nm Ti barrier layer was 176 cm^3^·s^−1^·cm^−2^, and its sorption quantity within 2 h was 289 Pa·cm^3^·cm^−2^. The results showed that, compared with the ZrCoCe getter film without a barrier layer, the binding force between the ZrCoCe getter film with a barrier layer and the Ge substrate was significantly improved. Therefore, the ZrCoCe getter films with the Ti barrier layers are more suitable for MEMS vacuum packaging devices in harsh environments, which is of great significance for the development of MEMS vacuum devices.

## Figures and Tables

**Figure 1 materials-16-02916-f001:**
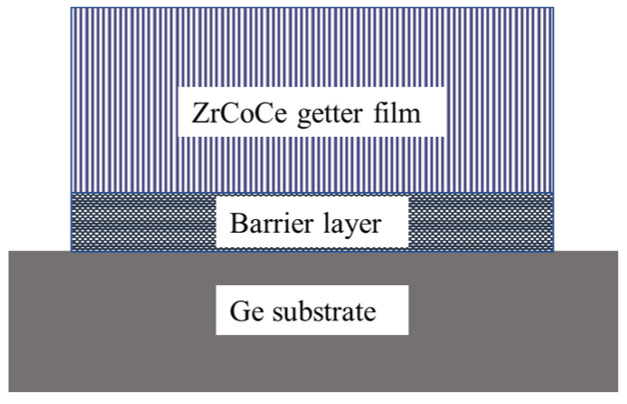
Structure of ZrCoCe getter films.

**Figure 2 materials-16-02916-f002:**
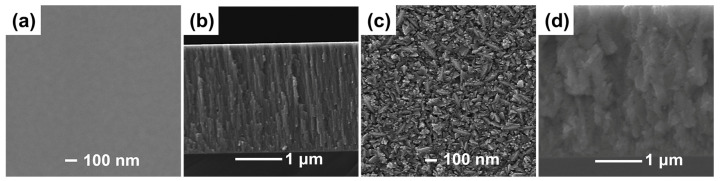
The surface and cross-sectional SEM images of (**a**,**b**) the ZrCoCe and (**c**,**d**) Ti barrier layers.

**Figure 3 materials-16-02916-f003:**
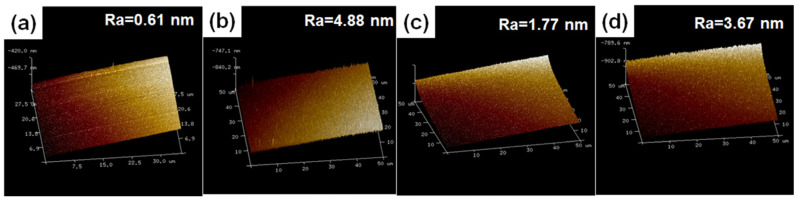
The surface roughness of the ZrCoCe barrier layer (**a**), Ti barrier layer (**b**), ZrCoCe/ZrCoCe getter film (**c**) and ZrCoCe/Ti getter film(**d**).

**Figure 4 materials-16-02916-f004:**
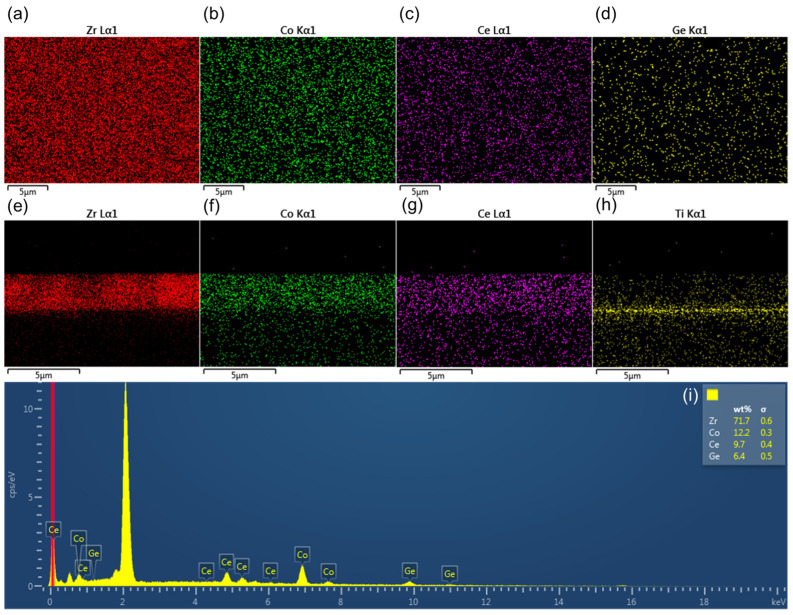
Energy Dispersive Spectroscopy surface images of ZrCoCe getter film (**a**) Zr, (**b**) Co, (**c**) Ce and (**d**) Ge; Energy Dispersive Spectroscopy cross-sectional images of ZrCoCe/Ti getter film (**e**) Zr, (**f**) Co, (**g**) Ce and (**h**) Ti; (**i**) Spectrum of ZrCoCe getter film.

**Figure 5 materials-16-02916-f005:**
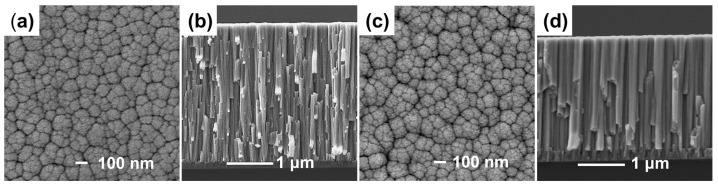
The surface and cross-sectional SEM images of (**a**,**b**) ZrCoCe/ZrCoCe and (**c**,**d**) ZrCoCe/Ti getter film.

**Figure 6 materials-16-02916-f006:**
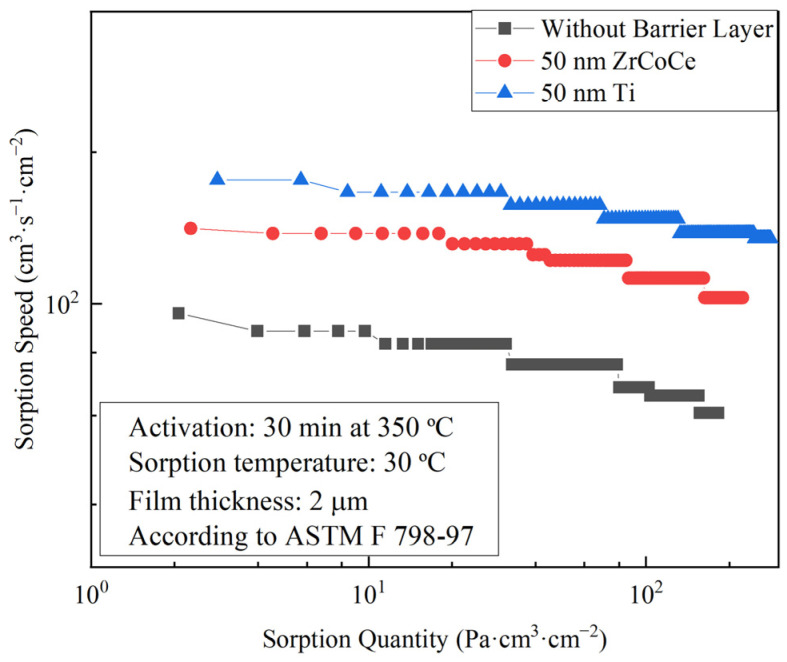
Sorption performance of ZrCoCe getter films with different barrier layers.

**Figure 7 materials-16-02916-f007:**
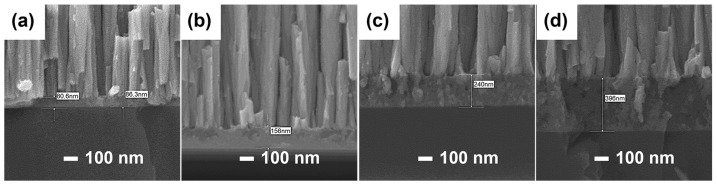
The cross-sectional micromorphology of the Ti barrier layer when the thickness is (**a**) 85 nm, (**b**) 156 nm, (**c**) 240 nm, and (**d**) 396 nm, respectively.

**Figure 8 materials-16-02916-f008:**
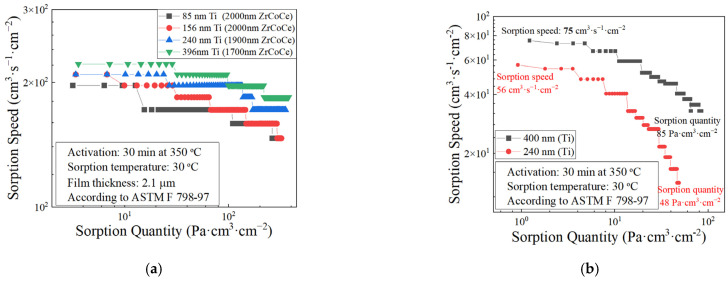
(**a**) Sorption performance of ZrCoCe getter films with different thicknesses of Ti barrier layers; (b)sorption performance of 240 nm and 400 nm Ti barrier layers.

**Figure 9 materials-16-02916-f009:**
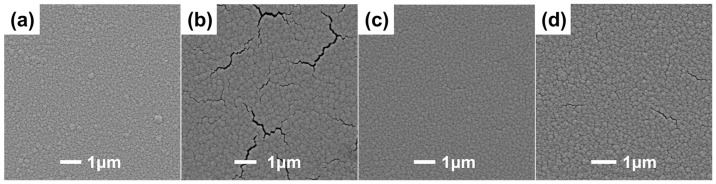
SEM images of the (**a**,**b**) ZrCoCe/ZrCoCe and the (**c**,**d**) ZrCoCe/Ti getter films after vibration and activation.

**Figure 10 materials-16-02916-f010:**
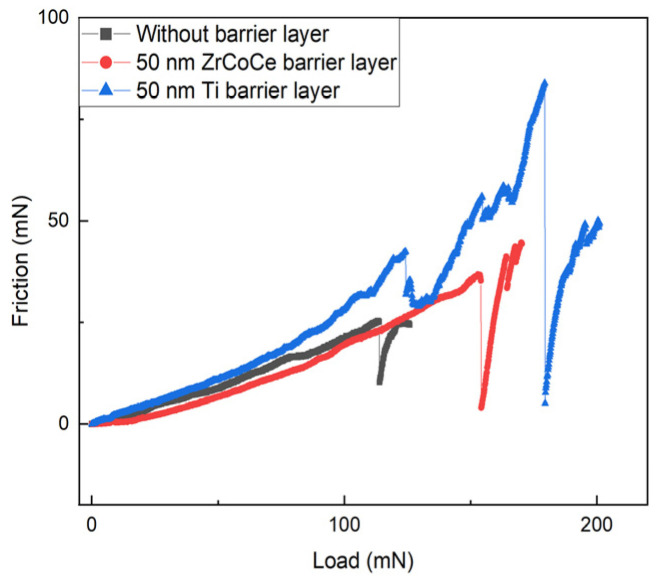
Binding force between Ge substrates and ZrCoCe getter film with different barrier layers.

**Table 1 materials-16-02916-t001:** Comparison of different gettering materials.

Gettering Materials	Activation Temperature/°C	Advantages and Disadvantages
TiZrV	180	Low activation temperature; oxides of V are toxic
ZrVFe	300~450	
ZrCoRE	250–450	Low activation temperature; environmental friendliness; MEMS compatibility

**Table 2 materials-16-02916-t002:** Magnetron sputtering process parameters.

Parameters	Barrier Layer		Getter Film
	Ti	ZrCoRE	ZrCoRE
Power supply	DC	DC	DC
Target to substrate distance/cm	7	7	7
Sputtering power/W	130	130	150
Deposition time/min	10	5	180
Sputtering Ar gas pressure/Pa	0.4	0.4	4.0

**Table 3 materials-16-02916-t003:** Data of sorption performance of ZrCoCe getter films [28].

Getter Film/2 μm	Initial Sorption Speed/cm^3^·s^−1^·cm^−2^	Sorption Quantity in 2 h/Pa·cm^3^·cm^−2^
Ti/ZrCoCe	176	289
ZrCoCe/ZrCoCe	141	223
ZrCoCe in this work	95	182
ZrCoCe in the literature	84	138
Pd/ZrCoCe	100	180

**Table 4 materials-16-02916-t004:** Data of sorption performance of ZrCoCe getter films with different barrier layers.

The Thickness of Ti/nm	Initial Sorption Speed/cm^3^·s^−1^·cm^−2^	Sorption Quantity in 2 h/Pa·cm^3^·cm^−2^
85	190	300
156	200	323
240	213	354
396	221	378

**Table 5 materials-16-02916-t005:** Binding force between ZrCoCe getter film with different barrier layers and Ge substrates.

Materials of Barrier Layer	Binding Force/mN	Materials of Barrier Layer
Without barrier layer	114	Without barrier layer
ZrCoCe	154	ZrCoCe
Ti	171	Ti

## Data Availability

The data presented in this study are available on request from the corresponding author. The data are not publicly available due to privacy.

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
