# Peer review of "Influence of Barrier Layers on ZrCoCe Getter Film Performance"

_materials, 2023, doi:10.3390/ma16072916_

Round 1

Reviewer 1 Report

The authors have produced an interesting manuscript on the improvement of the vacuum degree inside the vacuum device by means of different barrier layers. The manuscript needs further revision.

- A comparative table of the system proposed by the authors and what appears in the literature should be included in the manuscript.

- Further characterisation is needed, e.g. surface roughness, optical transmittance, adhesion, water vapour transmission rate or contact angle.

- The data of sorption performance have to be compared with further literature references and indicate the differences obtained between their results and those obtained by the authors.

- The references used are good, but it would be advisable to add some more current bibliographical references.

- A spelling check of the manuscript is needed.

If the authors make these changes, in order to improve the quality of the manuscript, it could be accepted by the reviewer.

Author Response

Dear Editor and Reviewers,

We appreciate your great work very much for handling our manuscript and the positive and constructive comments. These comments are quite valuable and very helpful for improving our manuscript. The manuscript has been carefully revised in accordance with all the comments, which is highlighted in red color in the revised manuscript. The point-by-point responses and revisions are elucidated beneath. 

If you have any questions, please do not hesitate to contact us.

Thank you for your kind consideration.

Sincerely yours

Xin Shi

Reviewer 2 Report

Comments on the paper: Influence of barrier layers on ZrCoCe getter film performance

Manuscript ID: materials-2313956

The authors describe the deposition of Ti and ZrCoCe used as barrier layers for getter films. The work is interesting and proposes a technological solution to improve the internal-vacuum level and stability of microelectromechanical system vacuum devices. Next are some points that authors should consider for the work publication.

Only 50% of bibliographic references correspond to publications from the last 5 years. It's necessary to increase up to at least 75% of the references of the previous 5 years related to the subject to show the novelty of the work.

The surface and cross-sectional studies shown in Figures 2 and 3 are adequate, as well as the analysis performed by SEM.

Authors must present studies related to elemental and phase analysis of the deposited films through the techniques such as EDX, XPS, and DRX.

The authors need to increase the theoretical discussion about the activity mechanisms as barrier layers and getter films of Ti and ZrCoCe.

Author Response

(The authors gave the same response as above.)

Round 2

Reviewer 1 Report

The authors have made the changes suggested by the reviewers. The manuscript is of sufficient quality to be accepted for publication.